# Investigation of the Effect of the Same Polishing Protocol on the Surface Roughness of Denture Base Acrylic Resins

**DOI:** 10.3390/biomedicines10081971

**Published:** 2022-08-14

**Authors:** Margarida Martins Quezada, Helena Salgado, André Correia, Carlos Fernandes, Patrícia Fonseca

**Affiliations:** 1Faculty of Dental Medicine (FMD), Universidade Católica Portuguesa, 3504-505 Viseu, Portugal; 2Centre of Interdisciplinary Research in Health, Faculty of Dental Medicine (FMD), Universidade Católica Portuguesa, 3504-505 Viseu, Portugal; 3Faculty of Engeneering (FEUP), Universidade do Porto, 4200-465 Porto, Portugal

**Keywords:** polymethyl methacrylate, CAD-CAM, acrylic resins, prosthodontics, surface properties

## Abstract

This investigation aims to determine the effect of the same polishing protocol on the surface roughness (Ra) of different resins obtained by different processing techniques. Acrylic resins obtained by CAD/CAM technology overcame the disadvantages identified in conventional materials. A total of thirty samples (six of each resin): self-cured, heat-polymerized, injection molded, CAD/CAM 3D-printed and CAD/CAM milled were prepared. JOTA^®^ *Kit* 1877 DENTUR POLISH was used to polish the samples by two techniques: manual and mechanized, with a prototype for guided polishing exclusively developed for this investigation. The Ra was measured by a profilometer. The values were analyzed using ANOVA, Games–Howell post-hoc test and One-sample *t*-test, with *p* < 0.05. Manual polishing produces lower values of Ra compared to mechanized polishing, except for injected molded resins (*p* = 0.713). Manual polishing reveals significant differences between the resin pairs milling/3D-printing (*p* = 0.012) and thermopolymerizable/milling (*p* = 0.024). In the mechanized technique only, significant differences regarding the R_a_ values were found between the self-cured/3D-printed (*p* = 0.004) and self-cured/thermopolymerizable pair resins (*p* = 0.004). Differences in surface roughness values can be attributed to the inherent characteristics of the resin and the respective processing techniques.

## 1. Introduction

Various materials have been used for denture base material, although the most commonly used has been poly(methyl methacrylate) (PMMA) [1,2,3]. According to the American Dental Association (ADA) Specification No. 12, prosthetic base polymers are classified into several types, depending on the polymerization activation reaction and their composition: type I (thermopolymerizable), type II (self-curing) and type III (thermoplastic) [4].

The widespread use of PMMA is due to its easy manipulation, reparation and polishing and is both esthetic and stable. Although, it has inherent disadvantages, such as polymerization shrinkage, allergies to residual monomer, poor wear resistance and low fracture and flexural strength [2]. One of the main disadvantages associated with PMMA obtained by conventional processing methods is the limitation of its physical and mechanical properties. Factors such as dimensional changes, susceptibility to fracture, the presence of residual monomer, increased risk of infections associated with prosthetic base and presence of porosity compromise the aesthetic results performance and hygiene, conditioning its survival [2,5].

With technological development, CAD/CAM (Computer-Aided Design/Computer-Assisted Machining) [6] technologies are being implemented in the dental field [5] for the fabrication of removable prostheses, provisional prosthetic rehabilitations and implant-supported prostheses in order to overcome the limitations of conventional methods. The CAM manufacturing process can occur by addition (rapid prototyping) or subtraction (computer numerical control machine—CNC) [7,8]. The additive method or 3D printing uses images from a digital file to create an object by depositing successive layers of the selected material. The subtraction method consists of removing material from a pre-formed block until the desired geometry is obtained [8].

Among the properties of the material used in the manufacture of dental prostheses, surface roughness is a factor of clinical relevance since it directly or indirectly affects the retention of microbial plaque in tissues in contact with the materials, increasing the risk of surface fatigue and decreasing their biocompatibility [9,10,11]. The threshold without statistical relevance to microbial colonization is 0.2 µm [12]. Above this value, pathological disorders, such as prosthetic stomatitis and oral candidiasis [13], can emerge as a result of microbial accumulation due to poor oral hygiene, abrasive processes during chewing and mechanical cleaning and reduced salivation associated with older patients. For that, mechanical and chemical polishing processes of prosthetic bases surface are indicated to reduce this accumulation [11].

The roughness of denture base materials is mainly affected by inherent material characteristics, the polishing technique and the individual operator’s skills [9,10]. Its characterization is performed using several parameters, although the average roughness (Ra) is the most used. By definition, Ra is the average value of the deviation of a measuring profile from a central line along the measuring length [14].

The purpose of this laboratory investigation is to determine the effect of the same polishing protocol on the surface roughness of different prosthetic base resins obtained by different processing techniques. The null hypothesis tested was that the surface roughness after polishing is indifferent to the type of acrylic resin processing used.

## 2. Materials and Methods

Thirty square samples (six samples from each type of resin) with 20 × 20 × 3 mm dimensions were divided into six groups: five groups containing five specimens of each type of acrylic resin and a control group containing one specimen of each type of resin.

The control group was not submitted to the polishing protocol. The study group was submitted to the JOTA^®^ *Kit* 1877 DENTUR POLISH (Jota AG, Rüthi, Switzerland) protocol, according to the manufacturer’s instructions by a manual technique and a mechanized technique.

### 2.1. Samples Preparation

Five types of commercially available denture base materials were tested: autopolymerized resin (AP) (Probase^®^ Cold; Ivoclar Vivadent, Liechtenstein), heat-polymerized resin (HP) (Probase^®^ Hot; Ivoclar Vivadent, Liechtenstein), injected molded resin (IM) (*iFlex^TM^*; tcs^®^, Signal Hill, CA), CAD/CAM 3D-printed resin (3D) (CediTEC DB; VOCO GmbH, Germany) and CAD/CAM milled resin (M) (*V—Print dentbase*; VOCO GmbH, Cuxhaven, Germany).

All samples were subjected to an evaluation of quality by analyzing the presence of fractures or irregularities on the surface detectable to the eye and by the verification of pre-defined dimensions with an analogic caliper (ROSTFREI GEHARTET caliper, Brütsch/Rüegger Tools Ltd., Urdorf, Switzerland).

### 2.2. Polishing Protocol

JOTA^®^ *Kit* 1877 DENTUR POLISH (Jota AG, Rüthi, Switzerland) was used to polish the samples. The manufacturer’s instructions advocate the use of a bullet-shaped drill for acrylic prosthetic bases. 

In both techniques, each drill (coarse grain, medium grain and fine grain) was used for 30 s at a speed of 5000 rpm by a micromotor-coupled handpiece (STRONG 206, SAESHIN^®^, Daegu, Korea).

To guarantee the same processing conditions for the specimens and to exclude their influence on the surface roughness, the samples used for the two polishing techniques were the same. The manual technique was applied on one side and the mechanized technique on the opposite side. This ruling was determined randomly. 

To ensure the repeatability of the polishing protocol between techniques, the sample was fixed in the same support in order to guarantee the parallelism with the bur. The polishing direction of the samples was maintained.

The polishing protocol was calibrated by a simple operator. The mechanized technique was applied with a prototype for guided polishing and the manual technique with the same device without the platform responsible for the fixation of the handpiece. 

#### Mechanized Technique—Prototype for Guided Polishing

Exclusively for this investigation, a device was developed in *Solidworks*^®^ (Figure 1) and impressed in Prusa mK3s by the method of fused deposition modeling (FMD) in acid polylactic (PLA).

The prototype for guided polishing is composed of two platforms. The first one is responsible for the fixation of the handpiece that allows the pressure exerted on the samples to be constant. The second platform allows the sample to be stabilized on a support with the pre-defined dimensions for samples in this study (20 × 20 × 3 mm). A track allows the movement of the first platform along the sample. The position of the handpiece relative to the base of the prototype ensures that the polishing of the surface of the sample is performed parallel to the support of the sample. 

With the use of this device, it is intended to standardize and control the polishing method, as it eliminates the variables related to the operator. 

### 2.3. Simulation of Oral Conditions

After polishing, samples were submersed in distilled water and placed in a stove (EHRET *BK 4106,* EHRET GmbH, Mahlberg, Germany) at a constant temperature of 37 °C for 24 h to simulate the rehydration of the acrylic for denture bases after the polishing protocol. 

### 2.4. Surface Roughness Measurement

The evaluation of surface roughness was performed using a contact profilometer (Hommelwerke with a linear unit LV-50 and a T8000 controller, Hommelwerke, Germany). A TK300 pickup with a vertical range of measurement of ±300 µm, a tip radius of 5 µm and a cone angle of 90° was used. Profilometric profiles were obtained with a traversing length of 4.8 mm. Using a cut-off length of 0.8 mm, the arithmetic mean roughness values (Ra) were extracted (Figure 2). On each specimen, three measurements were performed with an incremental distance of 1 mm between each reading line. The mean roughness value corresponding to the average of the three values was obtained. The measurement of each specimen was made in the direction perpendicular to the polishing direction.

### 2.5. Statistical Analysis

The R_a_ value obtained from the profilometer reading was then subjected to descriptive and inferential statistical analysis of the data performed using the IBM^®^ SPSS^®^
*version* 25.0 (IBM, EUA) with *p < 0.05*. The surface roughness values of the resins were represented through the mean, standard deviation and 95% confidence interval of the mean. A bivariate analysis of variance (ANOVA) was performed in order to compare the roughness distribution between the types of resins submitted to the same type of polishing. The Games–Howell post-hoc test was applied to determine the pairs of significant differences between the groups. The comparison of the average roughness values of the types of resins submitted to a given type of polishing with the respective controls was carried out using One-sample *t*-test. The Ra values were compared between different polishes using the *t*-test for independent samples.

## 3. Results

Table 1 represents the mean value of surface roughness, and standard deviation of the resins subjected to different polishing techniques, and Table 2 compares the distribution of surface roughness with the corresponding control resin.

Regarding the resins submitted to mechanized polishing, the roughness values in the 3D-printed (*p* = 0.004), injected molded (*p* = 0.003), and auto polymerized resins were significantly lower than the roughness values of the respective control resins. However, the distribution of surface values of the milled and injected molded resins was similar to the control group. The resins submitted to manual polishing showed significantly lower mean surface roughness values than the control resin (*p* < 0.001).

Table 3 shows the comparison of surface roughness values between the mechanized and manual techniques. All the resins submitted to manual polishing show surface roughness value significantly lower than resins submitted to mechanized polishing, except for injected molded resins (*p* = 0.713).

ANOVA analysis revealed significant differences in the distribution of roughness values between the groups of resins submitted to mechanized polishing (F = 3.323; *p* = 0.031). There were only significant differences regarding the roughness values between the auto polymerized and 3D printed resins (1.58 ± 0.19 vs. 0.86 ± 0.23; *p* = 0.004) and auto-polymerized and heat-polymerized (1.58 ± 0.19 vs. 1.07 ± 0.25; *p* = 0.004) (Table 4).

For resins subjected to manual polishing, the distribution of roughness values also differed significantly between resin types (F = 36.6; *p* < 0.001). Table 5 shows that there were significant differences between the pairs of milling resin/3D printed (*p* = 0.012) and heat polymerized/milling (*p* = 0.024). The average roughness value of the injected molded resin subjected to manual polishing differed significantly from all other resins.

## 4. Discussion

Conventional PMAA is the most used for prosthetic bases, although additives, such as polymerization initiators, accelerators, bonding agents and colorants, are added to the base composition, which influences the physical and chemical properties of these materials [15]. CAD/CAM technology has emerged in order to make the method of production uniform, reduce the production time and minimize the failure of the conventional method [2,15].

The surface roughness of prosthetic bases is an intrinsic physical characteristic of the materials that can be influenced by the operator’s manual skills, the polishing protocol and the composition of the material itself [5,9].

Regardless of the mechanical method being the most recommended for polishing protocols. As a result, secondary surface irregularities may emerge and act as niches for microbial retention [16]. In this study, it is possible to observe this phenomenon as both the Ra values for milled and injected molded resin test groups show higher values compared to the control group (0.98 ± 0.51 vs. 0.65 ± 0.09 μm and 1.44 ± 0.56 vs. 1.28 ± 0.37 μm, respectively). After the polishing protocol, the samples were submitted to distilled water at a constant temperature of 37 °C, which is recognized to reduce the residual monomer content produced as a result of an incomplete polymerization reaction that directly influences the surface roughness [17].

In resins subjected to manual polishing, the distribution of values differed significantly between resin types (F = 36.6; *p* < 0.001) and, also at the level of mechanized polishing, there were significant differences in the distribution of roughness values between resin groups (F = 3.323; *p* = 0.031). For that, when the same polishing protocol was applied, the difference in roughness values can be due to the inherent characteristics of the resin and respective processing techniques, so it is possible to reject the null hypothesis defined in this investigation.

In the group subjected to mechanized polishing protocol, there were significant differences regarding the roughness values between auto polymerized and 3D-printed resins (1.58 ± 0.19 vs. 1.07 ± 0.25 μm; *p* = 0.004). Lower Ra values of the 3D samples are described compared to conventional PMMA and are justified by the formation of air bubbles on the surface of self-curing resin compared to 3D printing technology that allows layer-by-layer polymerization [15,17] and prevents this production in the matrix [18]. There are also significant statistical differences between heat-cured and self-cured resins (1.58 ± 0.19 vs. 1.07 ± 0.25 μm; *p* = 0.004). The polymerization temperature and time affect the content of these resins. The polymerization reaction of heat-cured resin is activated by heat; in contrast, self-cured resin occurs by the addition of a chemical. Therefore, the degree of conversion of MMA monomer caused by chemical activation is not as high compared to thermal activation [2]. 

The distribution of Ra values for milled and injected molded resins was similar to the control group, in agreement with *Di* Fiore et al. [19], so the polishing protocol should be correctly selected to prevent higher surface roughness values. The current protocol did not produce an effect by reducing Ra values after polishing, which demonstrates that the polishing protocol may not be suitable for all types of prosthetic base resins. 

For resins subjected to manual polishing, the distribution of roughness values also differed significantly between the heat-cured and milling pair (*p* = 0.024). The pre-polymerized PMMA shows improvements in surface [15] compared to the heat-cured resin, mainly by the lower pressure condition, temperature and residual monomer levels [5]. Further, significant differences between pair of milled and 3D-printed resins were detected by Kraemer Fernandez et al. [20], similarly to the results obtained in the present study (*p* < 0.012).

The highest R_a_ value was obtained in the injected molded resin, similarly to the results of Gungor et al. [10], in which the value of these resins was significantly higher than heat-cured and self-cured (*p* < 0.001). However, in the study of Helal et al. [21], these resins, when compared to 3D-printed ones, showed lower values. It can be attributed to the pressure exerted during the processing process that prevents the creation of air bubbles. Further, the parameters of the impression of 3D-printed resins influence the surface morphology by the intermediate connections between layers. The highest values of Ra are associated with a 45° orientation of printing, which was used for this study. [17] Simultaneously, in contrast, the pressure exerted can generate a contraction in the polymerization that translates into surface porosity. In addition, compared to other resins, the internal structure of the resin is distinctive, as injected molded resins have a reduced number of binding agents that affect the hardness of the surface. They are formed by a linear chain of bon, whose resistance to surface pressure is diminished [21].

All resins subjected to manual polishing showed a mean roughness value significantly lower than resins subjected to mechanized polishing, except for injected molded resins (*p* = 0.713). Corsalini et al. [9] proposed a mechanized polishing method independent of the operator variability, whose aim would be to test the feasibility of a polishing protocol with repeatability of the operation. In this study, a comparison between manual polishing and mechanized polishing was performed. However, in contrast to the present investigation, the surface roughness value for manual polishing was globally higher than mechanized polished resins. Due to the similarity of polishing conditions between the prototype for guided polishing and the device used by Corsalini et al. [9,22], it can be extrapolated that there is a need for the calibration of this tool.

The present investigation presents several limitations. First, only three conventional, two CAD/CAM and one polishing kit were tested. In addition, the conditions in which the laboratory investigation was carried out do not allow the inherent changes in materials after long periods of time subjected to the conditions of the oral cavity to be mimicked. Further, the geometric form of the samples does not simulate the shape complexity of a prosthetic base. 

This investigation did not consider the impact of biological and mechanical properties on the durability of the resins used. 

Further studies should evaluate the effect of time on the inherent surface characteristics of acrylic base resins and compare different types of resins and systems of CAD/CAM technology.

## 5. Conclusions

Differences in surface roughness values between acrylic resins for denture bases can be attributed to the inherent characteristics of the material and to the processing method. CAD/CAM acrylic resins demonstrate lower values of Ra compared to the conventional PMMA. The present investigation creates new perspectives on a mechanized polishing method for the evaluation of surface roughness. 

## Figures and Tables

**Figure 1 biomedicines-10-01971-f001:**
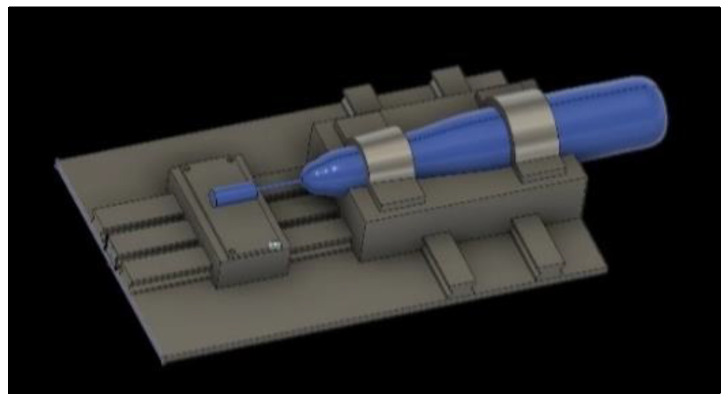
Solidworks^®^ design of the prototype for guided polishing.

**Figure 2 biomedicines-10-01971-f002:**
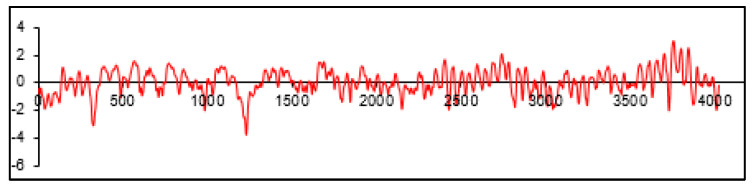
Graphic representative of profilometric measurements (µm).

**Table 1 biomedicines-10-01971-t001:** Mean and standard deviation of surface roughness (Ra—µm) according to the type of acrylic base resin submitted to different polishing methods.

Type of Resin	Mean	95% Confidence Interval of Mean		Standard Deviation
		Lower Bound	Upper Bound	
Mechanized Polishing				
3D	0.86	0.57	1.14	0.23
M	0.98	0.35	1.61	0.51
HP	1.07	0.75	1.38	0.25
AP	1.58	1.35	1.81	0.19
IM	1.44	0.74	2.14	0.56
Manual Polishing				
3D	0.52	0.48	0.55	0.03
M	0.29	0.19	0.40	0.08
HP	0.48	0.41	0.56	0.06
AP	0.50	0.30	0.70	0.16
IM	1.33	0.99	1.67	0.27

3D (3D-printed acrylic resin); M (milling acrylic resin); HP (heat polymerizing acrylic resin); AP (auto polymerizing acrylic resin); IM (injected molded acrylic resin).

**Table 2 biomedicines-10-01971-t002:** Comparison of surface roughness values (Ra—µm) between different types of acrylic resins submitted to mechanized and manual polishing and the corresponding control resin (One-sample *t*-test).

Type of Resin	Test	Control	Mean Deviation	Statistics	*p*-Value
Mechanized Polishing					
3D	0.86 ± 0.23	1.46 ± 1.01	−0.60	−5.854	0.004
M	0.98 ± 0.51	0.65 ± 0.09	0.33	1.472	0.215
HP	1.07 ± 0.25	1.77 ± 0.16	−0.70	−6.246	0.003
AP	1.58 ± 0.19	2.13 ± 0.11	−0.55	−6.570	0.003
IM	1.44 ± 0.56	1.28 ± 0.37	0.16	0.627	0.565
Manual Polishing					
3D	0.52 ± 0.48	1.46 ± 1.01	−0.94	−67.43	<0.001
M	0.29 ± 0.19	0.65 ± 0.09	−0.36	−9.700	<0.001
HP	0.48 ± 0.41	1.77 ± 0.16	−1.29	−46.60	<0.001
AP	0.50 ± 0.30	2.13 ± 0.11	−1.63	−22.56	<0.001
IM	1.33 ± 0.99	1.77 ± 0.16	−1.63	−22.56	<0.001

3D (3D-printed acrylic resin); M (milling acrylic resin); HP (heat polymerizing acrylic resin); AP (auto polymerizing acrylic resin); IM (injected molded acrylic resin).

**Table 3 biomedicines-10-01971-t003:** Comparison of surface roughness values (Ra—µm) between acrylic resin submitted to mechanized and manual polishing (*t*-test for independent samples).

Type of Resin	Mechanized Polishing	Manual Polishing	Statistics	*p*-Value
3D	0.86 ± 0.23	0.52 ± 0.48	3.265	0.029
M	0.98 ± 0.51	0.29 ± 0.19	3.002	0.037
HP	1.07 ± 0.25	0.48 ± 0.41	5.032	0.005
AP	1.58 ± 0.19	0.50 ± 0.30	9.856	<0.001
IM	1.44 ± 0.56	1.33 ± 0.99	0.386	0.713

3D (3D-printed acrylic resin); M (milling acrylic resin); HP (heat polymerizing acrylic resin); AP (auto polymerizing acrylic resin); IM (injected molded acrylic resin).

**Table 4 biomedicines-10-01971-t004:** Significance values of the bivariate comparison through the post-hoc Games–Howell test between the mean values of surface roughness (Ra—µm) of types of acrylic resins submitted to mechanized polishing.

Type of Resin	3D	M	HP	AP
M	0.983			
HP	0.658	0.997		
AP	0.004	0.231	0.040	
IM	0.325	0.678	0.678	0.977

3D (3D-printed acrylic resin); M (milling acrylic resin); HP (heat polymerizing acrylic resin); AP (auto polymerizing acrylic resin); IM (injected molded acrylic resin).

**Table 5 biomedicines-10-01971-t005:** Significance values of the bivariate comparison through the post-hoc Games–Howell test between the mean values of surface roughness (Ra—µm) of types of acrylic resins submitted to manual polishing.

Type of Resin	3D	M	HP	AP
M	0.012			
HP	0.803	0.024		
AP	0.999	0.206	0.999	
IM	0.011	0.003	0.008	0.005

3D (3D-printed acrylic resin); M (milling acrylic resin); HP (heat polymerizing acrylic resin); AP (auto polymerizing acrylic resin); IM (injected molded acrylic resin).

## Data Availability

Not applicable.

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
