# Peer review of "Investigation of the Effect of the Same Polishing Protocol on the Surface Roughness of Denture Base Acrylic Resins"

_biomedicines, 2022, doi:10.3390/biomedicines10081971_

Round 1

Reviewer 1 Report

This manuscript assesses in thirty samples (six of each resin) the effect of the same polishing protocol on the surface roughness of different denture base resins obtained by five different processing techniques (self-cured, heat-polymerized, injected molded, CAD/CAM 3D printed and CAD/CAM milled). The manuscript contains five keywords, two figures, five tables, and nineteen references. Overall, it is a correct and well-conducted manuscript, although some remarks are made in different sections.

Keywords
The manuscript presents five keywords. For keywords, where possible, please use Medical Subject Headings terms (MeSH Terms). Please, avoid the use of abbreviations as keywords. Strictly, none of them is a MeSH term. Nevertheless, some alternative MeSH terms proposed could be “polymethyl methacrylate” better than  “PMMA”, or “acrylic resins” rather than “acrylic base resin”. These suggestions about keywords are optional, not mandatory.

Materials and Methods
Page 3, line 130. Please, provide further information about the statistical software (company, address, etc.).
Was the significance level established with a p-value equal to or less than 0.05? Please clarify this.

Results
Tables 4 and 5 are not cited in the text. Please cite them. 

Discussion
Page 6, line 217. As specified in the reference list, is “Di Fiore et al.[16]”, not “Fiore et al.[16]”. Please, consider correcting this.
Page 7, line 226. According to the reference list, is “Kraemer Fernandez et al.[17]”, not “Fernandez et al.[17]”. Please, modify this.

References
Total number of the manuscript references: 19.
The reference format matches the journal's instructions about references. Nevertheless, in reference number 4, the commas after the semicolons should be removed according to the following reference format proposed for Journal Articles:
Author 1, A.B.; Author 2, C.D. Title of the article. Abbreviated Journal Name Year, Volume, page range.

Figures
Total number of the manuscript figures: 2.
The figures have appropriate figure legends.

Tables
Total number of the manuscript tables: 5.
In the tables, please consider adding footers explaining the abbreviations (SD in table 1, F in table 4) and Statistics (tables 2 and 3).
Table 5 has an appropriate footer. 

Author Response

Dear reviewer,  

After a period of analysis, the suggestions were accepted and alterations were introduced to the revised manuscript by section. 

Thank you for your appreciation.

Reviewer 2 Report

Thank you for giving me this opportunity to review the in vitro study article entitled, "Investigation of the effect of the same polishing protocol on the surface roughness of denture base acrylic resins".

I here carefully reviewed the submitted set of the manuscript and found it possibly merits of publication. However the major revisions should be mandatory to meet the scientific standard for publication, I'm afraid.

1. Why did you chose these five types of commercially available denture base materials, which are deferent in the type of polymerization. These should be also introduced in the Introduction section and should be discussed in the Discussion section.

2. "Polishing procedure" is not scientific. How did you verify the deviations of each technical differences in both manual and mechanical polishing techniques? The methods recruited in this study shouldn't be well enough scientifically. Repeated experiments by the same or various technical specialists for example should be evaluated and each technical distributions should be assessed to verify this experimental methods. Because this study is very sensitive issues evaluated and discussed.

3. The characteristics of  each type materials should be further discussed, especially for durability for both mechanical and biological functions. 

Further peer-review should be needed to re-evaluate the suitability for publication or not.

Author Response

Dear reviewer, 

All the comments have been taken into account.

Response to point 1: The selected prosthetic base resins were different in the type of polymerization, since the aim was to understand the influence of the processing techniques of the materials on the surface roughness.  For this purpose, the variables influencing roughness were excluded and the same polishing protocol was applied and the operator variables controlled by the prototype for guided polishing. Also, a paragraph has been introduced in the introduction regarding the American Dental Association (ADA) specification no. 12 which classifies polymers according to the polymerisation activation reaction and their composition and the disadvantages related to their use. Thus, it is justifiable to introduce new materials in the market, such as resins for prosthetic bases obtained by CAD/CAM technology, in which it is crucial to understand their properties, as surface roughness.

Response to point 2:  The term “polishing procedure” was substituted by “polishing protocol”. The methods established for this investigation can no longer be changed. The polishing protocol in both techniques was calibrated to be performed by only one operator in order to reduce operator-related biases. The deviations between techniques were reduced by ensuring that in both techniques the specimen was fixed in a support, that the handpiece speed was constant, that polishing was performed parallel to the surface and that the polishing time was the same. The only different characteristic between techniques was the digital pressure to which the specimen was subjected.

Response to point 3: It was introduced a phrase in the discussion that indicates that further studies should evaluate the influence of mechanical and biological functions on the durability of the materials.

Thank you for your appreciation.

Reviewer 3 Report

Dear Editor,

Current manuscript describes investigation of the effect of the same polishing protocol on the surface roughness of denture base acrylic resins. I think it can be published after minor revision.

1. Abstract section must be rephrased. Abstract must include obtained data.

2. Introduction section needs to improve. Please use of new references  between 2018 to 2022.

3. Please specify the country and company that made the used materials in this research.

4. Discussion of the manuscript must be deep with related references for supporting the sentences

5. Conclusion must be improved

6. Please compare the obtained results with similar work in this field in a separate table.

Author Response

Dear Reviewer,

All the comments have been taken into account.

Response to point 1: Data were included in the abstract. Abstract were rephrased.

Response to point 2: An update of the references was done.  The ones who are not included between 2018 – 2022 are referred to Specifications, Normative and some studies related to mechanized polishing that were not updated in this period of time.

Response to point 3: We add a comment indicating that the country and company are already indicated in this original manuscript.

Response to point 4: References were added to support the sentences, included in the interval suggested.

Response to point 5: Conclusion were rephrased.

Response to point 6: The results were already compared to the existing literature in the discussion.

Thank you for your appreciation.

Round 2

Reviewer 2 Report

Thank you for giving me this opportunity to re-review the revised manuscript.

I here carefully reviewed the re-submitted set of the manuscript and the revisions were well conducted on the concerns raised at the first round.